# Relationship between Self-Identity Confusion and Internet Addiction among College Students: The Mediating Effects of Psychological Inflexibility and Experiential Avoidance

**DOI:** 10.3390/ijerph16173225

**Published:** 2019-09-03

**Authors:** Kuan-Ying Hsieh, Ray C. Hsiao, Yi-Hsin Yang, Kun-Hua Lee, Cheng-Fang Yen

**Affiliations:** 1Department of Child and Adolescent Psychiatry, Kaohsiung Municipal Kai-Syuan Psychiatric Hospital, Kaohsiung 80276, Taiwan; 2Graduate Institute of Medicine, College of Medicine, Kaohsiung Medical University, Kaohsiung 80708, Taiwan; 3Department of Psychiatry and Behavioral Sciences, University of Washington School of Medicine, Seattle, WA 98195, USA; 4Department of Psychiatry, Children’s Hospital and Regional Medical Center, Seattle, WA 98195, USA; 5School of Pharmacy, Kaohsiung Medical University, Kaohsiung 80708, Taiwan; 6Department of Educational Psychology and Counseling, National Tsing Hua University, Hsin-Chu City 30013, Taiwan; 7Department of Psychiatry, Kaohsiung Medical University Hospital, Kaohsiung 80708, Taiwan

**Keywords:** internet addiction, PI, EA, self-identity

## Abstract

Internet addiction (IA) has become a major public health problem among college students. The aim of this study was to examine the relationship between self-identity confusion and IA and the mediating effects of psychological inflexibility and experiential avoidance (PI/EA) indicators in college students. A total of 500 college students (262 women and 238 men) were recruited. Their levels of self-identity were evaluated using the Self-Concept and Identity Measure. Their levels of PI/EA were examined using the Acceptance and Action Questionnaire-II. The severity of IA was assessed using the Chen Internet Addiction Scale. The relationships among self- identity, PI/EA, and IA were examined using structural equation modeling. The severity of self-identity confusion was positively associated with both the severity of PI/EA and the severity of IA. In addition, the severity of PI/EA indicators was positively associated with the severity of IA. These results demonstrated that the severity of self-identity confusion was related to the severity of IA, either directly or indirectly. The indirect relationship was mediated by the severity of PI/EA. Self-identity confusion and PI/EA should be taken into consideration by the community of professionals working on IA. Early detection and intervention of self-identity confusion and PI/EA should be the objectives for programs aiming to lower the risk of IA.

## 1. Introduction

### 1.1. Internet Addiction in College Students

Internet addiction (IA) has become a major global public health problem. Several studies have proposed diagnostic criteria for IA based on various concepts originally adopted from formal psychiatric disorders such as substance dependence [1] and pathological gambling [2] on the *Diagnostic and Statistical Manual of Mental Disorders, Fourth Edition (DSM-IV)* and impulse control disorder in the DSM-IV Text Revision [3]. Although IA has not been recognized as a disorder by the DSM-5 or the World Health Organization, the related diagnosis of internet gaming disorder and gaming disorder have been included in the Conditions for Further Study section of the DSM-5 [4] and the International Classification of Diseases, respectively. The common concepts of IA in previous studies include poorly controlled preoccupations, urges, withdrawal, tolerance, or behaviors associated with Internet use which lead to psychosocial distress and significant impairment at home, school, and work, as well as in health or interpersonal relationships [5,6,7]. The measures developed based on these concepts of IA can be used to evaluate the individuals’ severity of IA on the evidence base.

IA is predominant among young people, especially college students. College students normally have free and unlimited access to the Internet, flexible schedules, and are free from their parents’ interference. They use the Internet for studying, gaming, social networking, gambling, chatting, shopping, and watching pornographic videos [8,9]. Previous epidemiological studies revealed that the prevalence of IA in college students varied by country. Specifically, IA prevalence ranged from 3.7% in Japan [10] to 13.6% in China [11] and 15.3% in Taiwan [12]. Such deviation has been attributed to differences in study design, sampling, assessment tools, diagnostic criteria, and cultural backgrounds. IA has been associated with sleep disturbance [13], poor physical health [14], low self-esteem [15], poor academic performance [16], and mental health problems [17]. Therefore, IA is a crucial problem among college students and requires investigation.

Investigating the factors related to IA is the first step toward preventing its incidence and providing early treatment to improve mental health. IA is the result of interaction between individuals and their environment [18,19]. Several individual factors have been discovered as being related to IA. Research has demonstrated that depression and anxiety [20,21], a decreased sense of meaning in life [22], personality traits such as psychosis and bizarre mentation [23], low self-esteem [24,25], cognitive distortion [26], high impulsivity [27], and poor emotional regulation [28] are all associated with IA.

### 1.2. Self-Identify Confusion and IA

Self-concept formation, also termed personal identity formation, as a main developmental task in adolescence, includes the acceptance of physical changes and the development of social and emotional competencies and self-efficacy [29]. Previous studies have indicated significant deficits in self-concept formation among individuals with IA compared with those without [30,31]. Deficits in self-concept are characterized by low self-esteem; indecisiveness; inability to concentrate on required or suggested tasks; uncertainty regarding future objectives; an unclear description of self; problems in engaging in intimate relationships; and difficulties in social roles, values, and selections [29]. The Internet provides individuals with deficits in self-concept with a tempting environment to socially interact with others anonymously and to obtain success through Internet gaming or other such activities. These characteristics of the Internet increase the risk of IA in individuals with deficits in self-concept [32,33]. However, a lack of performance-related positive experiences on the Internet might also further deteriorate a self-concept deficit [31]. Therefore, investigating the relationship between self-identify confusion and IA is crucial to understand the mechanism of developing IA.

### 1.3. Psychological Inflexibility and Experiential Avoidance and IA

Psychological inflexibility and experiential avoidance (PI/EA) are defined as rigid strategies guided by psychological reactions and an unwillingness to experience unpleasant events or privations [34]. PI/EA contributes to the development, maintenance, and exacerbation of problematic behaviors [35], problematic substance use [36,37], and addictive behaviors [38]. PI/EA was reported as being positively associated with IA in one cross-sectional study [39].

### 1.4. Self-Identify Confusion and PI/EA

Research has determined three common types of self-identity confusion: Disturbed identity (a strong tendency to acquire the thoughts, feelings, beliefs, and problems of others in adulthood) [40], unconsolidated identity (failure to take on self-defining roles and demonstrate stable beliefs, attitudes, and values) [41], and lack of identity (sudden and dramatic shifts in self-image with respect to goals, values, vocational aspirations, sexual identity, and types of friends) [4]. People with self-identity confusion may experience difficulties accepting themselves and defusing cognitive stockades, and therefore, tend to develop PI/EA [42]. Some studies have revealed the therapeutic importance of cultivating psychological flexibility in the face of a self-identity crisis resulting from physical illness [43,44]. Throughout the process of accepting one’s current self, psychological flexibility is essential for reconstructing one’s self-concept, such that behaviors can be changed or redirected according to personal long-term values [44]. Although it is also possible that PI/EA may predate the maturity of self-concept, the present study focused on the influence of self-identity confusion on PI/EA and examined the mediating effects of PI/EA on the relationship between self-identity confusion and IA [45].

### 1.5. Aims of This Study

No study has examined the relationship between self-identify confusion, PI/EA, and IA. This study investigated the mediating effects of PI/EA on the relationship between self-identity confusion and IA among college students. We hypothesized that self-identity confusion is positively associated with IA and that PI/EA mediate the relationship between self-identity confusion and IA.

## 2. Materials and Methods 

### 2.1. Participants

Participants were recruited using an advertisement posted for college students aged between 20 and 30 years. Five hundred and three college students responded to the advertisement. Of them, three were excluded due to difficulties in understanding the meaning of research questionnaires. In total, 238 male and 262 female college students coming from 70 colleges located across Taiwan participated in this study. Their mean age was 22.1 years, with a standard deviation of 1.8 years. Written informed consent was obtained from all the participants prior to the assessment. The study was approved by the Institutional Review Board of Kaohsiung Medical University Hospital.

### 2.2. Measures

*Self-Concept and Identity Measure*. We used the Self-Concept and Identity Measure (SCIM) to assess the level of self-identity confusion [46]. The 27-item SCIM contains three subscales: Disturbed identity, unconsolidated identity, and lack of identity. Exploratory factor analysis revealed a three-factor structure of the SCIM in the college students. Confirmatory factor analysis validated the three-factor structure in the community sample. The correlations between the three subscales ranged from 0.32 to 0.53 [46]. Each item was rated using a seven-point Likert scale ranging from 1 (strongly disagree) to 7 (strongly agree). High total scores on the three subscales indicated tendencies for disturbed identity, unconsolidated identity, and lack of identity. The Cronbach’s α of the subscales in the present study ranged from 0.74 to 0.82.

*Chen Internet Addiction Scale*. We used the self-administered Chen Internet Addiction Scale (CIAS) to evaluate the participants’ severity of IA in the month preceding the study. The CIAS contains 26 items that are rated using a four-point Likert-type scale, with the total score ranging from 26 to 104 [47]. A higher total score indicated a more severe level of IA. The CIAS contains five subscales, including symptoms of compulsive use, withdrawal, tolerance, and problems in interpersonal relationships and health/time management. The Cronbach’s α of the five subscales of the CIAS in the present study ranged from 0.78 to 0.90. The 67/68 cutoff point of the CIAS is the optimal diagnostic cutoff point for Internet addiction in college students [48].

*The Acceptance and Action Questionnaire-II*. The Acceptance and Action Questionnaire-II (AAQ-II) [49] was revised from the original AAQ [50]. The AAQ-II consists of seven statements that represent various aspects of PI (e.g., “My painful experiences and memories make it difficult for me to live a life that I would value”) and EA (e.g., “I am afraid of my feelings”). The participants were asked to rate each of these statements on a scale of 1 (never true) to 7 (always true) based on their current experiences. A higher total score indicated a higher level of PI/EA. A study reported that the AAQ-II has adequate internal consistency and convergent and divergent validity [49]. The Cronbach’s α of the AAQ-II in the present study was 0.88.

### 2.3. Procedure and Statistical Analysis

Research assistants explained the procedures and methods for completing the research questionnaires to the participants individually. The participants were allowed to ask questions if they encountered problems while completing the questionnaires, and the research assistants helped them to resolve their problems.

The hypothesized model for the relationships among self-identity confusion, PI/EA, and IA is presented in Figure 1. Structural equation modeling (SEM) was used to estimate the parameters, test the model adequacy, and evaluate the extent of agreement between the observed data and the covariance matrix estimated from the model [51]. Amos version 18.0 software (IBM SPSS) was used to examine the goodness-of-fit index (GFI). The maximum likelihood method was used to analyze the data. In addition, the GFI, non-normed fit index (NFI), incremental fit index (IFI), comparative fit index (CFI), root mean square error of approximation (RMSEA), and standardized root mean square residual (SRMR) were used to evaluate the goodness of fit of the model [52]. On the basis of the goodness-of-fit requirement, the NFI, GFI, IFI, and CFI should be higher than 0.9; values of the RMSEA and SRMR lower than 0.05 are good; and values ranging between 0.05 and 0.09 are acceptable [52]. The Sobel test was applied to examine the mediating effect of PI/EA on the relationship between self-identity confusion and IA. A two-tailed p value of <0.05 was considered statistically significant.

### 2.4. Ethics

The study procedures were carried out in accordance with the Declaration of Helsinki. The Institutional Review Boards of Kaohsiung Medical University Hospital approved the study (ethical approval code number: KMUH-IRB-20120249; date of ethical approval: 30 August 2012). All participants were informed about the study and provided written informed consent. 

## 3. Results

In total, 85 (17%) participants were identified as having Internet addiction. The means, standard deviations, and correlation matrices of the measured variables are shown in Table 1. The results revealed significant correlations among the measured variables.

The goodness-of-fit indices of SEM for the hypothesized model on the relationships among self-identity confusion, PI/EA, and IA are listed in Table 2. The estimated coefficients of paths in the hypothesized model are presented in Figure 2. The results found that all NFI, GFI, IFI, and CFI values were higher than 0.9 and the SRMR value was lower than 0.05, which indicated the goodness of fit of the model was good. According to Hu and Bentler [52], the value of the RMSEA (0.085) was at the acceptable level. The result of the Sobel test confirmed the mediating effect of PI/EA on the relationship between self-identity confusion and IA (Z = 5.135, *p* < 0.05).

Moreover, all paths in the hypothesized model were significant. The severity of self-identity confusion was positively associated with the severity of IA and PI/EA. In addition, the severity of PI/EA was positively associated with the severity of IA. The severity of self-identity confusion was directly related to the severity of IA and indirectly related to the severity of IA through the increasing severity of PI/EA.

## 4. Discussion

This was the first study to examine the association between self-identity confusion, PI/EA, and IA. The results of this study revealed that self-identity confusion was directly related to IA and indirectly related to IA by the mediation of PI/EA. The results of previous studies on the relationship between self-identity styles and IA have been mixed. Some studies have demonstrated that normative styles of self-identity are protective factors of IA, whereas diffuse-avoidant styles are the risk factors of IA [53,54]. Several possible etiologies may account for the positive association between self-identity confusion and IA. First, the Internet provides college students who have self-identity confusion with an environment to explore their personal values, beliefs, and goals, and college students may use the Internet to escape the daily problems that they encounter which result from self-identity confusion. Second, according to the self-concept fragmentation hypothesis, college students with self-identity confusion may use their own different personalities to interact with others in online environments and feel more at ease within these various online environments than in the real world [55,56,57,58]. Third, young adults with self-identity confusion may seek peer advice from others online on issues related to developing self-identity during adolescence [59]. Although college students may use the Internet as a tool to develop their self-identity, IA may limit real-world interaction and therefore slow the development of self-identity [56]. In addition, exposure to different identities in online activities may exacerbate their self-identity confusion [60]. Therefore, college students require help with self-identity confusion to develop adaptive ways to interact with others and develop interpersonal relationships in order to improve their self-identity and consequently reduce their dependence on the Internet. Positive self-identity development could be promoted using Borba’s Esteem Builders Curriculum, which is one of the most comprehensive and widely used skills-based curricula [61]. Based on Marcia’s identity status theory [62], providing clarity of identity through fostering exploration creates the platform for identity commitment [63] and helps individuals become mature and competent during life transitions [64].

The present study discovered that PI/EA were significantly associated with IA in college students. The relationship between PI and IA has also been supported by the results of an imaging study that suggested that people with IA had difficulties in executive control and attention when switching tasks [65]. People with PI may experience psychosocial maladjustment, and stressful events [66] may cause them to use the Internet to avoid real-world stressors.

In this study, we found that PI/EA mediated the relationship between self-identify confusion and IA. A few possible mechanisms may account for the result. First, an unsuccessful formation of self-concept may lead individuals to experience self-identity diffusion, thereby increasing the risk of mental disorders, such as personality, depressive, and addictive disorders. These disorders usually persist into adulthood if they are not treated appropriately [29,63] and can compromise an individual’s self-concept. Persistent self-identity confusion may result in PI/EA [43,44] and result in psychological maladjustment [66]. The Internet may provide individuals with an opportunity to compensate for their self-directed negative feelings by providing opportunities for anonymous social interaction.

Second, people may experience various social-cognitive processes to construct and maintain their self-identity. Beronsky (1992) proposed three processing orientations that contribute to form and maintain a sense of self-identity: A procrastinating, diffuse-avoidant style; an open, informational style; and a conforming, normative style [67]. The individuals with a diffuse-avoidant identity style may tend to use the defense mechanism of avoidance, denial, suppression, and immaturity for masking painful inner conflicts [68,69]. Moreover, the individuals with the normative style may tend to use the defense mechanism of distorting, repressing, or denying reality that may limit self-awareness [70,71]. Therefore, the individuals with diffuse-avoidant or normative identity styles may develop PI/EA and are more likely to use the Internet than those with an informational identity style.

On the basis of our study results, we suggest that school counselors and mental health professionals routinely evaluate whether college students with IA have self-identity confusion and high PI/EA. For self-identity confusion, interventions that provide relational safety and support self-validity and self-exploration may be helpful in reducing the risk of IA [72].

The development of IA should be considered on the basis of ecological systems concepts [73]. Self-identity confusion is an individual factor related to IA. However, self-identity confusion may be influenced by sociocultural factors. Taiwanese culture based on Confucianism is more collectivistic-orientated than Western culture [74], and the difference in value orientation may make college students in Taiwan place greater importance on social relationships than those in Western societies. Whether the collectivistic-orientated culture may influence the development of self-identity and its relationship with IA warrants further study. It is also necessary to replicate the results of the present study in the individualistic-orientated societies.

Our study had a number of limitations. First, the cross-sectional research design limited our ability to draw conclusions regarding the causal relationship among self-identity confusion, PI/EA, and IA. Second, the study data were exclusively self-reported and may have therefore suffered from shared-method variance. Third, the participants in this study were college students who responded to the recruitment advertisement. Therefore, the results of this study might not be generalized to college students who did not participate in this study. Fourth, the present study used the AAQ-II to measure PI/EA. Although research determined that AAQ-II has adequate internal consistency and convergent and divergent validity [49], the items from the AAQ-II were found to be highly correlate with depression, anxiety, and stress, and therefore decreased its discriminant validity [75]. It is suggested that other measures, for example, the CompACT [76], might be needed in combination with the AAQ-II in assessing PI/EA.

Despite these limitations, this study contributes to the literature, as it is the first study to examine the relationship between self-identity confusion and IA and the mediating effects of PI/EA. Further research replicating this SEM model across different samples is suggested.

## 5. Conclusions

On the basis of our study, we found that self-identity confusion was directly related to IA and indirectly related to IA by the mediation of PI/EA. Therefore, self-identity confusion and PI/EA should be taken into consideration by the professionals working on prevention of IA. Moreover, we proposed that early detection and intervention for self-identity confusion may help college students to develop and consolidate self-identity, reduce PI/EA, and lower the risk of IA.

## Figures and Tables

**Figure 1 ijerph-16-03225-f001:**
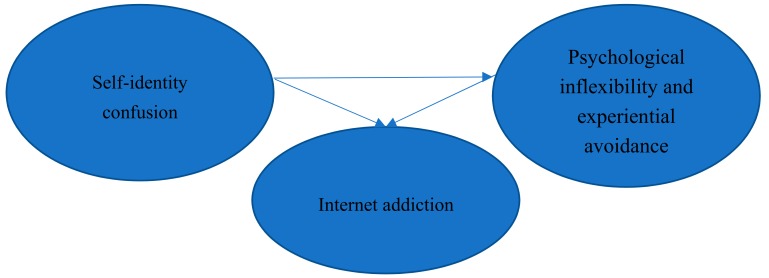
Hypothesized model of the associations among self-identity confusion, Internet addiction, and psychological inflexibility and experiential avoidance (PI/EA).

**Figure 2 ijerph-16-03225-f002:**
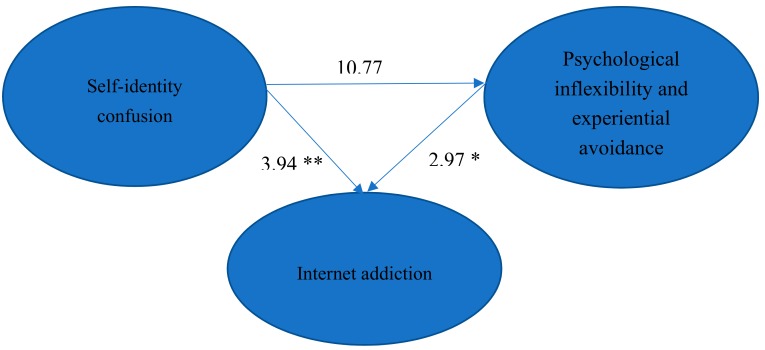
The estimated coefficients of the paths in the hypothesized model for the relationships among self-identity confusion, Internet addiction, and PI/EA. (* *p* < 0.05; ** *p* < 0.01).

**Table 1 ijerph-16-03225-t001:** The correlation matric of measured variables.

	Mean (SD)	1	2	3	4	5	6	7	8	9	10
1. Tolerance	8.9 (2.5)	1	0.66 *	0.67 *	0.57 *	0.70 *	0.26 *	0.28 *	0.26 *	0.24 *	0.30 *
2. Withdrawal	11.4 (3.3)		1	0.77 *	0.52 *	0.58 *	0.30 *	0.31 *	0.28 *	0.24 *	0.31 *
3. Compulsion	10.6 (3.1)			1	0.58 *	0.69 *	0.28 *	0.32 *	0.30 *	0.27 *	0.37 *
4. Time manage	10.5 (3.4)				1	0.68 *	0.20 *	0.27 *	0.23 *	0.21 *	0.29 *
5. Interpersonal	14.2 (4.2)					1	0.25 *	0.36 *	0.28 *	0.27 *	0.37 *
6. Psychological inflexibility	7.4 (3.3)						1	0.63 *	0.29 *	0.36 *	0.47 *
7. Experiential avoidance	9.8 (3.8)							1	0.35 *	0.39 *	0.52 *
8. Disturbed identity	33.7 (8.6)								1	0.24 *	0.56 *
9. Unconsolidated identity	18.4 (8.7)									1	0.52 *
10. Lack of identity	24.8 (8.4)										1

* *p* < 0.001.

**Table 2 ijerph-16-03225-t002:** The goodness-of-fit index of structural equation modeling for the hypothesized model.

Type	Goodness of Fit Index	The Full Model
Absolute fit indices	χ^2^	4.620
	df	32
	RMSEA	0.085 (*p* < 0.09)
	GFI	0.940 (*p* > 0.09)
Relative fit indices	NFI	0.941 (*p* > 0.09)
	IFI	0.953 (*p* > 0.09)
	CFI	0.953 (*p* > 0.09)
	SRMR	0.036 (*p* < 0.05)

χ^2^: chi-square; RMSEA: Root Mean Square Error of Approximation; GFI: Goodness-of-Fit Index; NFI: Non-normed-Fit Index; IFI: Incremental Fit Index; CFI: Comparative Fit Index; SRMR: Standardized Root Mean Square Residual.

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
