# Peer review of "Relationship between Self-Identity Confusion and Internet Addiction among College Students: The Mediating Effects of Psychological Inflexibility and Experiential Avoidance"

_ijerph, 2019, doi:10.3390/ijerph16173225_

Round 1

Reviewer 1 Report

This is a timely and valuable research article.  The investigation of ACT concepts such as experiential avoidance and psychological inflexibility and their relationship with disordered internet use is of significant merit.  Overall, I rate the manuscript highly.  However, there are several issues that I think need to be addressed to improve the manuscript and potential impact of the research. 

The authors did outline the Internet Addiction as a concept, but I think that the conceptualisation is insufficient.  I appreciate the attempt to maintain brevity and reduce verbosity.  However, I think given the context of vociferous debate in the literature regarding the parameters of the Internet Addiction concept, it is not possible to avoid spending time carefully articulating the researchers’ position in this debate.  In other words, I think the authors need to spell out what they perceive Internet Addiction to be.  For example, I acknowledge the authors used the Chen Internet Addiction Test (CIAT) which I accept to be of sufficient quality in terms of psychometric properties.  However, although the instrument refers to the phenomenon as an ‘Addiction’, it is not obvious from the arguments in the introduction that authors share this conceptualisation.  The clinical disorder of addiction is used rather loosely in the ‘internet disorder’ literature, and it can be difficult for authors to be precise in their language given the misuse of the label throughout the literature.  It may seem pedantic, however I think the readers must carefully specify how they perceive the disorder, and whether they do indeed believe it constitutes a psychiatric disorder.  And if they do, I think it prudent to justify it, to at least some extent.

I think there also needs to be some justification of the measures used.  The authors do a good job in articulating the psychometric properties and merits of the instruments applied, but there are multiple alternatives instruments that could have been used for both PI/EA and IA.  For example, current literature questions whether the AAQ II should be replaced with the compACT.  It would be valuable to understand why the authors chose to remain with the AAQ II.

Both Figure 1 and Figure 2 are of insufficient quality.  There are spelling errors and the aesthetics need to be improved e.g. alignment etc.

I think the introduction needs to make a stronger case of why under-developed or unstable self-concept causes psychological inflexibility or experiential avoidance.  It is not that I believe this direction is not possible, rather it is the case that the alternative direction may also be plausible.  Development of experiential avoidance begins in early childhood (often from as young as 4 years old), so it is possible that cognitive inflexibility and experiential avoidance may pre-date adolescence, where self-concept matures and develops.  Put simply, I think the authors must support their hypothesised direction outlined in Figure 1.

I think the RMSEA result requires further clarification beyond it being acceptable.  For example, under current accepted guidelines the RMSEA of the model would not meet the requirements for a ‘good fit’.  This may not be entirely obvious to all of the readers, therefore for transparency, further elaboration on the quality of fit indices would be valuable.

I was taken aback by the lack of consideration of cultural variation in the theoretical concepts and more so with respect to the implications proposed.  Given that the findings from a Taiwanese population are being contrasted against literature from multiple, variant cultures, can be automatically integrate these disparate studies.  For example, there is theoretical discussion about the development or self-identity, self-concept and identity.  Is this suitable; are there no cultural variations across the studies.  Perhaps there are no cultural variations, however I think the article needs to refer to this point, as it may a question in the readers’ minds, as they consider applying any of the findings.

Finally, I think the suggested recommendations are vastly premature. I entirely appreciate that the findings of the study are impressive, and suggestive of strong external validity.  However, I think it is common practice in SEM studies to replicate the model across multiple samples before attempting such grandiose conclusions and recommendations.

Author Response

August 11, 2019

Prof. Dr. Paul B. Tchounwou
Editor-in-Chief
International Journal of Environmental Research and Public Health

Revised Manuscript: IJERPH-553849

Title: Relationship Between Self-Identity Confusion and Internet Addiction Among College Students: The Mediating Effects of Psychological Inflexibility and Experiential Avoidance

Dear Professor Tchounwou:

We are grateful for the valuable comments from the editors and reviewers on our manuscript. We would like to thank the reviewers for the considering our manuscript interesting and gave us many information. The following responses have been prepared to address all of the reviewers’ comments in a point-by-point fashion. Please let us know anything else we should provide.

Thank you very much.

Yours sincerely

Cheng-Fang Yen

Department of Psychiatry, Kaohsiung Medical University Hospital,

100 Tzyou 1st Rd, Kaohsiung, Taiwan

Tel: +886-7-3124941

Fax: +886-7-3134761

Kun-Hua Lee, PhD

Department of Educational Psychology and Counseling, National Tsing Hua University, 521 Nan-Da Rd., Hsin-Chu City, 30014, Taiwan

Tel: (+886) 3-5715131 ext 73815

Fax: (+886) 3-5252205

For Reviewer 1

Comment 1

The authors did outline the Internet Addiction as a concept, but I think that the conceptualisation is insufficient.  I appreciate the attempt to maintain brevity and reduce verbosity.  However, I think given the context of vociferous debate in the literature regarding the parameters of the Internet Addiction concept, it is not possible to avoid spending time carefully articulating the researchers’ position in this debate.  In other words, I think the authors need to spell out what they perceive Internet Addiction to be.  For example, I acknowledge the authors used the Chen Internet Addiction Test (CIAT) which I accept to be of sufficient quality in terms of psychometric properties.  However, although the instrument refers to the phenomenon as an ‘Addiction’, it is not obvious from the arguments in the introduction that authors share this conceptualisation.  The clinical disorder of addiction is used rather loosely in the ‘internet disorder’ literature, and it can be difficult for authors to be precise in their language given the misuse of the label throughout the literature.  It may seem pedantic, however I think the readers must carefully specify how they perceive the disorder, and whether they do indeed believe it constitutes a psychiatric disorder.  And if they do, I think it prudent to justify it, to at least some extent.

Response

We appreciate your valuable comment. We added introductions of concepts of Internet addiction (IA) listed below to the first paragraph of Introduction section.

“Several studies have proposed diagnostic criteria for IA based on various concepts originally adopted from formal psychiatric disorders such as substance dependence [1] and pathological gambling [2] on the Diagnostic and Statistical Manual of Mental Disorders, Fourth Edition (DSM-IV) and impulse control disorder in the DSM-IV Text Revision [3]. Although IA has not been recognized as a disorder by the DSM-5 or the World Health Organization, the related diagnosis of internet gaming disorder and gaming disorder have been included in the Conditions for Further Study section of the DSM-5 [4] and the International Classification of Diseases, respectively. The common concepts of IA in previous studies include poorly controlled preoccupations, urges, withdrawal, tolerance, or behaviors associated with Internet use which lead to psychosocial distress and significant impairment at home, school, and work, as well as in health or interpersonal relationships [5–7]. The measures developed based on these concepts of IA can be used to evaluate the individuals’ severity of IA on the evidence base.” Please refer to line 40 to 52.

Comment 2

I think there also needs to be some justification of the measures used. The authors do a good job in articulating the psychometric properties and merits of the instruments applied, but there are multiple alternatives instruments that could have been used for both PI/EA and IA. For example, current literature questions whether the AAQ II should be replaced with the compACT. It would be valuable to understand why the authors chose to remain with the AAQ II.

Response

Thank you for your suggestion. Research did find that the items from the AAQ-II highly correlated with depression, anxiety, and stress (Tyndall et al., 2019) and suggested that the CompACT (Frances et al., 2016,) should be incorporated with the AAQ-II in assessing psychological inflexibility. We added the discussion listed below into the limitation s of Discussion section.

“Forth, the present study used the AAQ-II to measure PI/EA. Although research determined that AAQ-II has adequate internal consistency and convergent and divergent validity [50], the items from the AAQ-II were found to be highly correlate with depression, anxiety, and stress and therefore decreased its discriminant validity[76]. It is suggested that other measures, for example, the CompACT [77] might be needed to incorporate with the AAQ-II in assessing PI/EA.” Please refer to line 269 to 274.

Comment 3

Both Figure 1 and Figure 2 are of insufficient quality. There are spelling errors and the aesthetics need to be improved e.g. alignment etc.

Response

Thank you for your suggestion. We’ve modified both Figure 1 and Figure 2 as suggested.

Comment 4

I think the introduction needs to make a stronger case of why under-developed or unstable self-concept causes psychological inflexibility or experiential avoidance. It is not that I believe this direction is not possible, rather it is the case that the alternative direction may also be plausible. Development of experiential avoidance begins in early childhood (often from as young as 4 years old), so it is possible that cognitive inflexibility and experiential avoidance may pre-date adolescence, where self-concept matures and develops. Put simply, I think the authors must support their hypothesised direction outlined in Figure 1.

Response

Thank you for your suggestion. In the revised manuscript we added a new paragraph as below to strengthen the hypothesized direction between self-identity confusion and psychological inflexibility or experiential avoidance.

“1.4. Self-identify confusion and PI/EA

Research has determined three common types of self-identity confusion: disturbed identity (a strong tendency to acquire the thoughts, feelings, beliefs, and problems of others in adulthood) [40], unconsolidated identity (failure to take on self-defining roles and demonstrate stable beliefs, attitudes, and values) [41], and lack of identity (sudden and dramatic shifts in self-image with respect to goals, values, vocational aspirations, sexual identity, and types of friends) [42]. People with self-identity confusion may have difficulties in accepting themselves and defusing cognitive stockades and therefore tend to develop PI/EA [43]. Some studies have revealed the therapeutic importance of cultivating psychological flexibility in the face of a self-identity crisis resulting from physical illness [44,45]. Throughout the process of accepting one’s current self, psychological flexibility is essential for reconstructing one’s self-concept such that behaviors can be changed or redirected according to personal long-term values [45]. Although it is also possible that PI/EA may predate the maturity of  self-concept, the present study focused on the influence of self-identity confusion on PI/EA and examined the mediating effects of PI/EA on the relationship between self-identity confusion and IA [46].” Please refer to line 91 to 105.

Comment 5

I think the RMSEA result requires further clarification beyond it being acceptable.  For example, under current accepted guidelines the RMSEA of the model would not meet the requirements for a ‘good fit’. This may not be entirely obvious to all of the readers, therefore for transparency, further elaboration on the quality of fit indices would be valuable.

Response

We revised the description as below for the quality of fit indices to make it more transparent.

“The results found that all NFI, GFI, IFI, and CFI values were higher than 0.9 and the SRMR value was lower than 0.05, which indicated the goodness of fit of the model was good. According to Hu and Bentler [53], the value of the RMSEA (0.085) was at the acceptable level.” Please refer to line 182 to 185.

Comment 6

I was taken aback by the lack of consideration of cultural variation in the theoretical concepts and more so with respect to the implications proposed.  Given that the findings from a Taiwanese population are being contrasted against literature from multiple, variant cultures, can be automatically integrate these disparate studies.  For example, there is theoretical discussion about the development or self-identity, self-concept and identity.  Is this suitable; are there no cultural variations across the studies.  Perhaps there are no cultural variations, however I think the article needs to refer to this point, as it may a question in the readers’ minds, as they consider applying any of the findings.

Response

We agree that it is important to consider the influence of sociocultural background on the development of self-identity and Internet use behaviors. In the revised manuscript we added a new paragraph as below in Discussion section to address it.

“The development of IA should be considered on the basis of ecological systems concepts [74]. Self-identity confusion is an individual factor related to IA; however, self-identity confusion may be influenced by sociocultural factors. Taiwanese culture based on Confucianism is more collectivistic-orientated than Western culture [75], and the difference in value orientation may makes college students in Taiwan place greater importance on social relationships than those in Western societies. Whether the collectivistic-orientated culture may influence the development of self-identity and its relationship with IA warrants further study. It is also necessary to replicate the results of the present study in the individualistic-orientated societies.” Please refer to line 256 to 263.

Comment 7

Finally, I think the suggested recommendations are vastly premature. I entirely appreciate that the findings of the study are impressive, and suggestive of strong external validity.  However, I think it is common practice in SEM studies to replicate the model across multiple samples before attempting such grandiose conclusions and recommendations.

Response

Thank you for your comment. In the revised manuscript we modified our conclusion as below.

“On the basis of our study, we proposed prevention and intervention programs for IA in college students should consider self-identity confusion and PI/EA. Moreover, early detection and intervention for self-identity confusion may help college students to develop and consolidate self-identity confusion may lower the risk of IA.” Please refer to line 279 to 282.

Reviewer 2 Report

Many thanks for sending me this paper to review: internet addiction (IA) is a growing phenomenon and problem, although with significant variation in rates across different countries. This paper examines the mediating effects of psychological inflexibility and experiential avoidance on its relationship with self-identity confusion.

General comments:

A.      Both the title and the abstract were, on the whole, well written and reflect the content of the paper proper. I have a concern with the last sentence (see conclusion below).

B.      The introduction gives a good overview of the literature and where this study fits into this.

C.      The methods give a detailed overview of the study.

1.       Cross-sectional study

2.       Participants were recruited using self-reported questionnaires - a non-clinical population.

3.       The measures and the statistical analysis were appropriate and well described.

D.      The results give a brief description of the findings. There is no description of the sample, or of how the participants scored on the various measures used. In line one, of Results, the authors state that the “means, standard deviations and correlation matrices of the measured variables are shown in table 1”. Unfortunately the means and SDs are not reported. Also the reader is given no information on the degree of severity of IA, or indeed what proportion of the population meet the criteria for this on the CIAS.

E.       The discussion provides a reasonable summary of this study, and covers the area of comparison between this study and the existing literature well. The limitations are discussed in reasonable detail, however it omits the recruitment bias inherent to it methodology. The recommendation for ACT as a treatment measure is not supported by the evidence generated in this paper.

F.       The conclusion in my view goes a little too far in its recommendations which are not adequately supported by these findings.

G.     The references are profuse, relevant and up to date.

H.      On the whole the tables and figures, are clear. However regarding Figure 2 , the legend states at the bottom what is meant by values denoted *, but not for those denoted **

I. Minor typographic errors.

Author Response

August 11, 2019

Prof. Dr. Paul B. Tchounwou
Editor-in-Chief
International Journal of Environmental Research and Public Health

Revised Manuscript: IJERPH-553849

Title: Relationship Between Self-Identity Confusion and Internet Addiction Among College Students: The Mediating Effects of Psychological Inflexibility and Experiential Avoidance

Dear Professor Tchounwou:

We are grateful for the valuable comments from the editors and reviewers on our manuscript. We would like to thank the reviewers for the considering our manuscript interesting and gave us many information. The following responses have been prepared to address all of the reviewers’ comments in a point-by-point fashion. Please let us know anything else we should provide.

Thank you very much.

Yours sincerely

Cheng-Fang Yen

Department of Psychiatry, Kaohsiung Medical University Hospital,

100 Tzyou 1st Rd, Kaohsiung, Taiwan

Tel: +886-7-3124941

Fax: +886-7-3134761

Kun-Hua Lee, PhD

Department of Educational Psychology and Counseling, National Tsing Hua University, 521 Nan-Da Rd., Hsin-Chu City, 30014, Taiwan

Tel: (+886) 3-5715131 ext 73815

Fax: (+886) 3-5252205

For Reviewer 2

Comment

I have a concern with the last sentence (see conclusion below). The conclusion in my view goes a little too far in its recommendations which are not adequately supported by these findings.

Response

Thank you for your comment. In the revised manuscript we modified our conclusion as below.

“On the basis of our study, we proposed prevention and intervention programs for IA in college students should consider self-identity confusion and PI/EA. Moreover, early detection and intervention for self-identity confusion may help college students to develop and consolidate self-identity confusion may lower the risk of IA.” Please refer to line 279 to 282.

Comment

There is no description of the sample, or of how the participants scored on the various measures used. In line one, of Results, the authors state that the “means, standard deviations and correlation matrices of the measured variables are shown in table 1”. Unfortunately the means and SDs are not reported. Also the reader is given no information on the degree of severity of IA, or indeed what proportion of the population meet the criteria for this on the CIAS.

Response

Thank you for your comments.

We described sex distribution and mean age in Methods, “2.1. Participants” section.

“In total, 238 male and 262 female college students coming from 70 colleges located across Taiwan participated in this study. Their mean age was 22.1 years, with a standard deviation of 1.8 years.” Please refer to line 116 to 118.

We apologized for our omitting the means and standard deviations of the variables in Table 1. We added them into Table 1. In the revised manuscript we added the proportion of the participants who meet the criteria for this on the CIAS.

“The 67/68 cutoff point of the CIAS is the optimal diagnostic cutoff point for Internet addiction in college students [49].” Please refer to line 137 to 138.

“In total, 85 (17%) participants were identified as having Internet addiction.” Please refer to line 175.

Comment

The limitations are discussed in reasonable detail, however it omits the recruitment bias inherent to it methodology. The recommendation for ACT as a treatment measure is not supported by the evidence generated in this paper.

Response

Thank you for your reminding. We added it as one of the limitations of this study as below.

“Third, the participants in this study were college students who responded to the recruitment advertisement. Therefore, the results of this study might not be generalized to college students who did not participate in this study.” Please refer to line 267 to 269.

We also deleted the recommendation for ACT as a treatment measure in the revised manuscript.  

Comment

On the whole the tables and figures, are clear. However regarding Figure 2, the legend states at the bottom what is meant by values denoted *, but not for those denoted **

Response

Thank you for your reminding. We modified the legend of Figure 2 into “* p < 0.05; ** p < 0.01”.

Comment

Minor typographic errors

Response

Thank you for your reminding. We have examined the manuscript thoroughly and corrected the typographic errors.
